# Review of the Mechanisms of Snake Venom Induced Pain: It’s All about Location, Location, Location

**DOI:** 10.3390/ijms23042128

**Published:** 2022-02-15

**Authors:** Vance G. Nielsen, Michael T. Wagner

**Affiliations:** 1Department of Anesthesiology, University of Arizona College of Medicine, Tucson, AZ 85719, USA; 2Department of Anesthesiology, New York Medical College, Valhalla, NY 10595, USA; michael.wagner@wmchealth.org

**Keywords:** neurotoxicity, venomous snake bite, acute pain, chronic pain, phospholipase A_2_, serine protease, metalloproteinase, snake venom peptides

## Abstract

Pain—acute, chronic and debilitating—is the most feared neurotoxicity resulting from a survivable venomous snake bite. The purpose of this review is to present in a novel paradigm what we know about the molecular mechanisms responsible for pain after envenomation. Progressing from known pain modulating peptides and enzymes, to tissue level interactions with venom resulting in pain, to organ system level pain syndromes, to geographical level distribution of pain syndromes, the present work demonstrates that understanding the mechanisms responsible for pain is dependent on “location, location, location”. It is our hope that this work can serve to inspire the molecular and epidemiologic investigations needed to better understand the neurotoxic mechanisms responsible for these snake venom mediated diverse pain syndromes and ultimately lead to agent specific treatments beyond anti-venom alone.

## 1. Introduction

A primordial fear shared by humankind is often heralded by the sudden onset of intense pain in a limb—the beginning of envenomation and neurotoxicity from a snake bite. This is only the onset of what may be a complex experience composed of fear, injury to limb and potentially loss of life. However, in the maelstrom of this event, the cause and nature of the painful experience associated with envenomation may be as varied as the biochemical and proteomic composition of the venom [1,2,3] to which the bitten are exposed. Acute local pain [4,5] can spread systemically, experienced as headache [6,7,8,9,10,11,12,13,14,15,16], eye pain [17,18,19,20,21,22,23,24,25,26,27,28], chest pain [27,28,29,30,31,32,33,34], focal back pain [34,35], abdominal pain [5,33,36,37,38,39,40,41,42,43,44,45,46] or generalized pain [34,38,39] that may last only during the immediate episode or can progress into chronic pain syndromes such as migraine headache or complex regional pain syndrome (CRPS) [7,47,48,49,50,51,52]. Given the complex and seemingly unpredictable outcome in the matter of snake venom induced pain, it would be important to understand the molecular mechanisms underlying this experience that subsequently dictate appropriate treatments.

## 2. Location: Molecular Mechanisms of Venom Mediated Pain

### 2.1. Overview

The first paradigm of location concerns the molecular site of action that causes pain. The molecular composition of venom obtained from snakes can be remarkably complex, composed of biogenic amines, enzymes, peptides, and other substances to incapacitate their prey [1,2,3,53,54,55,56,57,58,59,60,61,62,63,64,65,66,67,68,69,70,71,72]. The reader is referred to a few recent excellent reviews for greater detail [57,58,59,62]. Examples of such compounds include proteins, small molecular weight, non-enzymatic compounds, serine proteases, metalloproteinases, phospholipase A_2_, and 3-finger toxins [1,2,3,53,54,55,56,57,58,59,60,61,62,63,64,65,66,67,68,69,70,71,72]. These venom components or their enzymatic byproducts interact with a variety of receptors on Aδ and C pain fibers as recently reviewed [58,59,60]. The following subsections outline the likely mechanisms by which these venom components inflict pain in a myriad of ways.

The paradigm invoked by the aforementioned is that the venom proteins presented do in fact inflict pain either directly by stimulating specific receptors by binding to the receptors or locally generating compounds that bind, remotely by stimulating receptors distant from the bite site, or indirectly by causing pain syndromes as a result of ischemia or relentless muscle activation. While the administration of antivenom does attenuate pain syndromes in areas remote from the bite site by binding to and neutralizing circulating and locally presented venom compounds, tissue edema and relative tissue ischemia surrounding the bite site may prevent attenuation of pain secondary to antivenom not being able to be delivered to the envenomation strike point. Individual classifications of venom compound and molecular site of action are subsequently presented.

### 2.2. Small Molecular Weight, Non-Enzymatic Compounds (Direct Effects)

This category includes compounds such as biogenic amines (e.g., histamine, serotonin), kinins, eicosanoids, and other peptides that bind to their specific receptors [53,54,55,56,57,58,59,60]. These are found as preformed substances in venoms to varying degrees [53,54,55,56,57,58,59,60] and likely contribute to the initial pain sensations after being bitten. Most of these substances are rapidly metabolized in the envenomed tissues, circulation, or lung by enzymes such as histaminase, monoamine oxidase, 15-hydroxyprostaglandin dehydrogenase, and cytochrome 450s. Thus, without ongoing generation, the aforementioned preformed small molecular weight compounds would be expected to contribute to pain experienced at the bite site but not at distant sites, and only for a brief time. An exception could be non-enzymatic proteins, such as those in the venom of *Echis coloratus*, that activate the transient receptor potential vanilloid 1 (TRPV1) channel [60], which could leave the bite site without being degraded to cause pain elsewhere in the victim. Nevertheless, and critically, the vast majority of venomous snake bites are not as severe initially as they are in the minutes or hours that follow envenomation [4]. When considered as a whole, it is likely that the formation of these and other compounds via catalysis of the envenomed tissues by enzymes contained in venom contributes to progressively increasing local pain. However, as the subsequently described venom proteins are released into the circulation and cause pain in distant organs in a syndromic fashion, it should be remembered that as end-organ inflammation increases, so does release of the aforementioned small molecular weight compounds that may contribute to pain systemically. In summary, while preformed substances in venom likely cause early pain, it is the subsequently described enzyme classes contained in venom that contribute to pain at the bite site and in locations distant from the initial strike point.

### 2.3. Phospholipase A_2_ (PLA_2_) (Direct Effects)

While these snake venom enzymes are perhaps most feared for their properties as preganglionic neurotoxins (β-neurotoxins) that inflict apneic death [1,57,58,59], they also cause edema, tissue injury and, critically, pain [57,58,59,62,72,73,74,75,76]. PLA_2_ activity catalyzes phospholipids in the bite site and beyond after release into the circulation, resulting in formation of bradykinin, biogenic amines, prostaglandins, and other compounds that inflict pain [57,58,62,72,73,74,75,76]. For example, PLA_2_ isolated from *Crotalus durissus* species venom has been demonstrated to activate C fibers, resulting in the release of substance P, mast cell degranulation, and finally, release of histamine and serotonin [63]. A similar release of substance P and bradykinin was observed following the use of a secretory PLA_2_ isolated from *Naja mocambique mocambique* venom in a model of acute pancreatitis [64]. In the same vein, a PLA_2_ isolated from *Micruruus lemniscatus*, Lemnitoxin, was found to be a potent agent that degranulated mast cells [65]. A PLA_2_ isolated from *Bothrops atrox* venom, BatroxPLA_2_, caused release of IL-6 and formation of prostaglandin E_2_ (PGE_2_), leukotriene B_4_ (LTB_4_), and cysteinyl leukotrienes (CysLTs) in mice [66]. Further, snake venom PLA_2_ from *Bothrops* species also inflict pain via cellular release of adenosine triphosphate and potassium [75,76]. Of interest, a heteromeric toxin composed of a PLA_2_ with minimal enzymatic activity with Kunitz-like protein (MitTx) purified from the venom of *Micrurus tener tener* that activates acid-sensing ion channels (ASICs) independent of enzymatic activity has been identified as a source of pain [61]. Of equal importance, the role of PLA_2_ in the development of pain has been demonstrated by inhibition of these enzymes, which results in a decrease in pain in vivo [77,78]. For the interested reader, a more in-depth consideration of PLA_2_ is recommended [62]. Thus, it is likely that PLA_2_ significantly contribute to the pain syndromes subsequently presented.

### 2.4. Serine Proteases (Direct and Indirect Effects)

This class of snake venom enzyme is perhaps most notorious for inflicting coagulopathy following snake bite [1]; however, these enzymes are also demonstrated to contribute to pain in more than one manner. Serine proteases activate protease-activated receptor 2 (PAR2), which in turn generates pain in several settings [79,80]. Using murine models, human cancer cells secrete serine proteases that inflict pain when injected into the hind paw, and this pain was reduced with serine protease inhibitors [79]. In another investigation, the pain caused by injection of mice paws with formalin, bradykinin, or PAR2-activating peptide was reduced in animals with PAR2 deletion [79]. As for an example with snake venom, serine proteases purified from the venom of *Bothrops pirajai* significantly contributed to hyperalgesia in a murine paw bending model [67]. A second mechanism by which serine proteases may inflict pain is by causing regional arterial thrombosis via activation of coagulation [1], which would result in regional ischemic pain. Examples of ischemic pain will be presented in detail in the following sections. In summary, serine proteases likely play a significant role in envenomation associated pain.

### 2.5. Metalloproteinases (Direct and Indirect Effects)

Metalloproteinases also have a variety of proven or possible mechanisms by which they may contribute to snake bite pain, and there are several examples found in the literature. A metalloproteinase purified from *Bothrops atrox*, Batroxase, caused release of IL-6 and formation of PGE_2_, LTB_4_, and CysLTs in mice [66,71]. Further, metalloproteinases contained in *Bothrops jararaca* venom enhanced hyperalgesia in a murine model [68], as did a purified metalloproteinase, BaP1, contained in *Bothrops asper* venom, via TNF-α and PGE_2_-dependent mechanisms [69]. A final example is the hyperalgesic effect of a metalloproteinase, BpirMP, in a rat model that was purified from the venom of *Bothrops pirajai* [70]. As for other mechanisms, these enzymes have been associated with neuropathic pain, with cleavage of interleukin-1β resulting in the activation of microglial cells or astrocytes, depending on the metalloproteinase involved [81]. Further, similar to serine proteases, metalloproteinases are capable of activating PAR2 [82]. Lastly, this class of enzyme can exert potent procoagulant activity, resulting in arterial thrombosis and ischemic pain [1].

### 2.6. Fasciculins (Indirect Effects)

Fasciculins, found in *Dendroaspis* species (mambas) venom, are a subclass of three-finger toxins that exert their toxicity by causing uncontrollable fasciculations of skeletal muscle and subsequent paralysis and apneic death [2]. In addition to paralysis, fasciculations are painful, and continuous fasciculation can result in significant muscle damage and pain after recovering from the snake bite despite mechanical ventilation and pharmacological neuromuscular blockade [83]. Fasciculins bind to circulating acetylcholinesterase and inactivate the enzyme, allowing continuous exposure of the post synaptic membrane of neuromuscular junctions to acetylcholinesterase, resulting in fasciculations [84]. Similar pain, but to a far lesser degree, is observed postoperatively in muscular patients after administration of succinylcholine during the conduct of anesthetic induction [85]. This medication briefly (1–2 min) depolarizes skeletal muscle to effect temporary paralysis to facilitate endotracheal intubation, and the musculature is observed briefly to fasciculate [86]. Therefore, it is not surprising that patients that survive a mamba bite may complain of significant muscular pain afterwards [83]. Thus, fasciculins are a unique indirectly acting, pain-provoking agent in snake venom.

A diagrammatic and simplified summary of this section is provided in Figure 1. For a detailed review of the cellular and molecular mechanisms of pain, the interested reader is referred to an excellent review [86].

## 3. Location: Sites of Pain after Envenomation

### 3.1. Overview

The second paradigm of location concerns the anatomical site or organ affected by pain following snake bite. The site of pain in reference to the bite can be classified as either local (e.g., at or adjacent to the bite site) or remote. It can be postulated that if antivenom treatment attenuates pain in a remote site such as the abdomen, then the pain is being caused by the elaboration of small molecular weight compounds (e.g., biogenic amines, kinins, cyclooxygenase eicosanoids, etc.) derived from snake venom enzymatic activity (e.g., PLA_2_) that is inactivated by the antivenom as opposed to irreversible damage of neural structures by such enzymes.

### 3.2. Bite Site and Surrounding Tissue Pain

As mentioned in Section 2.2, preformed compounds contained in snake venom would be expected to cause initial pain at the bite site. Subsequently, the activity of PLA_2_, serine proteases and metalloproteinases most likely contribute to ongoing and continuous pain at and surrounding the bite site as presented in Section 2.3, Section 2.4 and Section 2.5. While snake venom constituents are released into the systemic circulation to cause some of the subsequently described pain symptoms/complexes, it should be noted that tissue edema at the bite site caused by these enzymes [1] may prevent adequate delivery of antivenom that would be expected to attenuate bite site pain. The primary impediment to this therapy is that antivenom antibody/antibody fragments are macromolecular and may have difficulty accessing target molecules in edematous tissue or at the bite site [87]. It should also be noted that the pattern of onset, duration, and severity of pain very much depends on the composition of the venom which is very species dependent [4]. The remainder of this section will address other pain syndromes caused by local application or systemic distribution of snake venom.

### 3.3. Eye Pain

Venom mediated eye pain (venom ophthalmia) has been extensively reported after various *Naja* species (spitting cobras) project their venom into their victim’s eyes [17,18,19,20,21,22,23] or after systemic envenomation by *Atractaspis* snake bite on rare occasion (remote, direct mechanism) [10]. In the case of *Atractaspis* envenomation, the eye pain occurred when the eyes moved, but no true ophthalmia was observed, and symptoms abated over time without any specific treatment [10]—thus, it is difficult to speculate what molecular mechanisms were at play. Of interest, too, Crotalid venom inadvertently projected into the eye of a patient that was striking the snake’s head caused severe ophthalmia [24], and nuchal gland secretions of the poisonous snake, *Rhabdophis tigrinus formosanus*, caused severe ophthalmia when they sprayed into the patient’s right eye [25]. Eye pain would be expected following ocular exposure to venom, given the presence of preformed compounds that inflict pain (see Section 2.2). Further, the enzymatic action of PLA_2_ and other enzymes on the cell membranes of the cornea and conjunctiva would be expected to amplify venom mediated pain and inflict edema and tissue injury as reported [17,18,19,20,21,22,23,24,25]. Of interest, Asian and African spitting cobras appear to have a greater percentage of PLA_2_ in their venom than do the corresponding non-spitting cobra species; further, it appears that the African spitting cobras have a greater percentage of cytotoxic 3-finger toxins in their venom than the Asian spitting cobras [28]. In general, venom ophthalmia has been treated successfully with local interventions: flushing with water, and administering a variety of medications including local anesthetics, antibiotics, antihistamines, steroids, and occasionally antivenom eye drops [17,18,19,20,21,22,23,24,25]. Interestingly, antivenom administered locally or systemically has not been demonstrated to be superior to non-specific treatments alone [26]. Further, systemic signs of envenomation do not occur in this setting [17,18,19,20,21,22,23,24,25]. From a molecular standpoint, once the venom induces edema, its enzymatic constituents are prevented from being adsorbed as evidenced by the lack of symptoms of systemic envenomation; and, conversely, the macromolecular antibodies of antivenom are not able to access the venom enzymes inflicting pain on the surface of the eye as documented by the lack of efficacy of antivenom administration. In conclusion, it is likely that the local, direct mechanisms responsible for venom ophthalmia involve preformed, small molecular weight compounds initially followed by enzymatic activity damaging the corneal and conjunctival cells that in general is limited in duration with appropriate therapy.

### 3.4. Headache

Acute headache can be experienced to a variable extent (11.6–100%) after snake bite on a distant body part, making this a remote form of venom-mediated pain [6,8,9,10,11,12,13,14,15,16]. In these cases, descriptions of the headache are lacking in detail, varying from simply noting headache without further comment or mentioning it as intense or focal (e.g., frontal headache) [6,8,9,10,11,12,13,14,15,16]. As a rule, with or without antivenom therapy, the acute headaches seem to resolve and generally are not the most remarkable issue facing envenomed patients [6,8,9,10,11,12,13,14,15,16]. On the other hand, the typical acute headache associated with envenomation does not persist after administration of antivenom [6,8,12,13], which strongly suggests that one or more of the aforementioned enzymes which would be inactivated by antivenom are responsible for the headache. Another barrier to narrowing down possible mechanisms includes a lack of description of the headache for the most part in these works to allow classification of it as muscular or vascular in nature. Thus, a great deal more must be done to better identify the particular type of acute headache that is caused by which venom enzyme.

Of interest, migraine headache that did not exist prior to the snake bite can become a chronic condition (5.6% of victims assessed), with features such as photophobia, faintness, and vertigo [7]. In general, of the type of snake that could be identified, bites from members of the families Viperidae and Elapidae were responsible for 47.8% of long-term headaches, implying some sort of permanent neurological damage caused by venom enzymes [7]. Surprisingly, 2 of 46 patients that developed migraine headache after being bitten had nonvenomous snakes involved, which is potentially indicative of a significant psychological etiology perhaps secondary to the trauma of being bitten at all [7]. In summary, acute and chronic headache can occur as a pain syndrome after a snake bite to a remote part of the body, and a great deal of investigation is needed to determine direct or indirect molecular mechanisms responsible.

### 3.5. Chest Pain

The scenarios of chest pain following a snake bite are more varied than the preceding pain syndromes [27,29,30,31,32,33,34], and the incidence is difficult to identify as most citations are case reports or limited case series. Chest pain following snake bite can appear as classic angina pectoris with electrophysiological and biochemical evidence of myocardial infarction [27,29,30,31] or be nonspecific in nature [33,34]. Of interest, infarction may be secondary to coronary thrombosis in areas of vessel stenosis [29], or in patients with vessels without atherosclerosis [31] or within drug-eluting stents maintaining the patency of coronary arteries [32]. Further, this myocardial damage may also occur in patients without coronary artery pathology as part of a hypersensitivity reaction that causes temporary coronary artery spasm, Kounis syndrome, associated with snake bite [30]. As can be seen, chest pain following venomous snake bite is variable in nature.

The mechanisms responsible for these clinical scenarios are remote, and treatment with antivenom when available has been either the primary or secondary remedy in addition to the several medical and catheterization interventions required to stop the chest pain [27,29,30,31,32,33]. The mechanism for pain is most likely ischemia secondary to coronary artery spasm or thrombosis [27,29,30,31]; however, how the spasm or thrombosis that occurs may have little to do with the venom, have to do with focal prothrombotic effects of the venom in the coronary artery or stent, or have to do with mast cell mediated focal coronary spasm. In the scenario wherein snake venom is not involved in ischemic injury, it could be the psychological stress with concordant increases in heart rate and blood pressure that may rupture an atherosclerotic plaque in a coronary artery that indirectly leads to thrombosis and myocardial ischemia. In the case of prothrombotic serine proteases or metalloproteinases contained in the venom, focal thrombi may form and therefore the effect would be considered an indirect mechanism. Kounis syndrome following venomous snake bite would also be mediated by indirect mechanisms that may be elicited by any of the components of the venom (peptides, enzymes) that would trigger a hypersensitivity response. Finally, as for the nonspecific chest pain experienced after venomous snake bite [32,33], the chest pain is usually part of a generalized pain experienced throughout the body and is not accompanied by electrophysiological or biochemical signs of myocardial ischemia. The pain resolves with antivenom [32] or without it [33], which is most likely a sequence of events consistent with a direct mechanism, such as serine proteases, metalloproteinases or PLA_2_ localizing to the peripheral nerve endings, nerves, or ganglia that would be accessible to antivenom administration. In conclusion, multiple remote, direct and indirect molecular mechanisms are responsible for chest pain after envenomation, but further investigation will be required to further define them.

### 3.6. Abdominal Pain and General Systemic Pain

Abdominal pain following venomous snake bite [5,33,36,37,38,39,41,42,43,44,45,46] can be remarkably variable secondary to the venom (0% [40] to 91% [41]) and can occur after administration of antivenom [40]. The three families of venomous snake, Colubridae, Elapidae, and Viperidae, can all inflict severe abdominal pain [5,34,36,37,38,39,41,42,43,44,45,46]. The onset of pain can be within a few minutes to hours, last hours to days, and pain generally resolves over time or with antivenom administration [5,34,36,37,38,39,41,42,43,44,45,46]. The abdominal pain usually involves painful cramping, nausea, and/or diarrhea [5,34,36,37,38,39,41,42,43,44,45,46]. In one case of severe rattlesnake envenomation, in the presence of severe and recurrent hypotension, constant abdominal pain was found to be secondary to ischemic and infarcted bowel, with death ensuing shortly thereafter [88]. Given the variable onset in pain, it is likely that small molecular weight peptides are responsible for remote, early pain by either directly stimulating the nervous system innervating the bowel or by indirectly causing excess motility or spasm of smooth muscle within the bowel which would be perceived as pain. Later onset abdominal pain would likely be caused by the direct effects of venom enzymes such as PLA_2_, serine proteases or metalloproteinases on intestinal tissue pain receptors or other locations within the innervation of the gut. Discerning what components of snake venom inflict abdominal pain is a field of investigation that is ripe for exploration.

Severe general systemic pain after snake bite [34,38,39], in most or all body cavities and limbs, occasionally occurs and usually resolves with antivenom or supportive therapy. It is likely that small molecular weight peptides or enzymes such as PLA_2_, serine proteases or metalloproteinases are responsible given the remote nature of onset and offset of pain, which is likely a direct effect of the proteins on tissue pain receptors or other locations within the nervous system. In addition, as mentioned previously, mamba venoms containing fasciculins can cause remote pain indirectly by causing painful muscular fasciculations in skeletal muscle throughout the body [2,83]. As with abdominal pain, the precise molecular mechanisms responsible for severe generalized pain after snake bite remain to be defined.

### 3.7. Back Pain

Back pain following venomous snake bite [34,35] that is focal is remarkably rare in the literature. Both cases involved one species of snake, *Proatheris superciliaris* (Lowland Swamp Viper; East African Lowland Viper; Peter’s Viper; Eyebrow Viper; Floodplain Viper), with one patient that was a collector located in the Czech Republic [34] and the other, a professional wildlife photographer and collector living in France [35]. The native location of this snake in south-western Tanzania and coastal plain of Mozambique. Both patients developed severe lumbar pain, followed by marked renal failure requiring hemodialysis [34,35]. Of interest, thrombocytopenia with only mild to moderate changes in plasmatic coagulation were noted with standard laboratory tests [34,35]. Neither patient was treated with specific antivenom as none exists. Neither report indicated when the lumbar pain abated. After several days, the first patient did not require dialysis and after several weeks, recovered [34]. However, the second patient was found to have suffered bilateral renal infarctions resulting is a need for hemodialysis until his kidney transplant 18 months later [35]. Given the clinical findings of these cases, this remote syndrome of severe back pain was most likely mediated by an indirect mechanism involving focal thrombosis caused by proatherocytin, a 34-kDa selective platelet protease activated receptor-1 (PAR1) agonist, a potent renal vasoconstrictor as previously reviewed [89]. The infarction mediated pain experienced by both patients likely abated as renal vasoconstriction decreased or ischemic tissue died. In conclusion, the infrequent onset of severe lumbar pain following snake bite has been associated with just one species, and is most likely secondary to a remote, indirect molecular mechanism involving severe renal vasoconstriction to the point of infarction [34,35,89].

### 3.8. Complex and Chronic Pain Syndromes

Complex and chronic pain syndromes following venomous snake bite are uncommon but debilitating [7,47,48,49,50,51,52]. These conditions include migraine headache or complex regional pain syndrome (CRPS). CRPS, as recently reviewed [90], is a syndrome of ongoing, chronic pain, disproportionate to the original injury (or may be idiopathic), involving peripheral nerve injuries. Three families of venomous snake, Colubridae, Elapidae, and Viperidae, can cause this type of neurotoxicity [47,48,49,50,51,52]. The affected area may exhibit hyperesthesia or allodynia, edema, hair loss, and temperature changes compared to the contralateral body part, and many of these changes are attributed to dysregulation of the sympathetic nervous system [90]. CRPS treatments are multimodal, involving medications such as steroids, non-steroidal anti-inflammatory agents, regional nerve blocks, and physical therapy [90]. While the cause of CRPS involves nerve injury, CRPS that occurs secondary to venomous snake bite can be considered both local and remote, as the bite site displays the aforementioned symptoms, but so do proximate parts of the limb bitten [47,48,49,50,51,52]. The nerve injury would be considered direct, and most likely involve acute neurotoxicity from enzymes such as PLA_2_ and metalloproteinases that damage tissues structurally and not just physiologically in an easily reversible manner (by antivenom). However, it is the indirect and complex mechanisms of the onset of chronic pain that is the neurotoxicity that may last from weeks to years despite therapy [47,48,49,50,51,52]. While the precise molecular mechanisms responsible for how snake venom injures peripheral nerves to the point of causing CRPS and migraine are not well defined, it is likely the improvements in acute management of snake bite may potentially decrease the incidence of this disabling neurotoxicity. A diagrammatic and simplified summary of this section is provided in Figure 2.

## 4. Location: Nature of Venom Mediated Pain as a Function of Geographic Origin of the Snake—“It’s Not Where the Snake Bites You, but Where You Are Bitten, That Matters”

This last section addresses the matter of identifying the pain syndromes most likely to be found in patients based on the location they are bitten in or the origin of the snake that bites a collector. The molecular mechanisms responsible for the aforementioned pain syndromes are driven by the proteome of the venom of the offending snake, and these can be remarkably diverse. The differences in the proteomes of venoms found in spitting cobras in Africa and Asia serve as an example [28]. As for examples of differences in pain syndromes based on geographical location, references to venom associated acute headache were readily found in Africa, Australia, Europe, and South America [6,8,9,10,11,12,13,14,15,16], but not elsewhere. Chest pain that is ischemic is observed in North America [29] and Asia [27,31,32] via thrombosis or via an allergic coronary compromise in Europe [30]; however, chest pain in Southeast Asia is not ischemic [33]. References to abdominal pain after venomous snake bite were found in Africa, Asia, and Europe [35,36,37,38,40,41,42,43,44,45]. Some pain syndromes are very regional, such as ophthalmia from spitting cobras in Africa or Asia [17,18,19,20,21,22,23] or focal back pain after a bite from a specific viper from Africa [34,35], or generalized muscular pain after being bitten by a mamba via fasciculins [83,84]. Lastly, as previously mentioned [4], pain from the bite site is almost always painful, covering snakes on nearly all continents (e.g., rattlesnakes and corral snakes in North, Central, and South America). From both a molecular and clinical point of view, the mechanism(s) responsible for neurotoxicity is very dependent on the evolutionary and geographical forces that result in the specific venom proteome responsible for the pain syndrome suffered. While it would be ideal to have a map of the specific pain syndromes inflicted by specific species, such an endeavor is not possible secondary to the lack of detailed (in terms of pain) epidemiological studies available concerning the thousands of species of venomous snakes that exist worldwide. Such an undertaking should be considered by the stakeholders in this field during the conduct of future investigation. We enclose a simplified illustration that summarizes this section as Figure 3.

## 5. Conclusions

The purpose of this brief review was to raise awareness of what little is known about the molecular mechanisms of snake venom responsible for pain, arguably the most debilitating neurotoxicity following snake bite. While venom induced paralysis and death by asphyxia is one of the most feared outcomes of envenomation, it is the hours, days, weeks, or years of pain that can follow a bite that leaves the greatest societal impact in terms of suffering and disability. Using our “location, location, location” paradigm to understand venom induced pain, we hope that the readership will appreciate the remarkably complex interactions of peptide and enzyme that result in novel neurotoxicity. We summarize this paradigm with the adjoining Figure 4. Not all venoms contain the same constituents; not all sensory neurons or other components of the nervous system are vulnerable to the same peptide or enzyme; not all tissues and organs have the same innervation or vulnerability to venom constituents; and, lastly, snakes have incredibly diverse venom proteomes, a diversity driven by geographical and other environmental factors. Documentation of specific pain syndromes in greater detail in future epidemiological studies of snake bite is also critical. In conclusion, it is our hope that this work can serve to inspire the molecular investigation needed to better understand the neurotoxic mechanisms responsible for these snake venom mediated diverse pain syndromes and ultimately lead to agent specific treatments beyond antivenom alone.

## Figures and Tables

**Figure 1 ijms-23-02128-f001:**
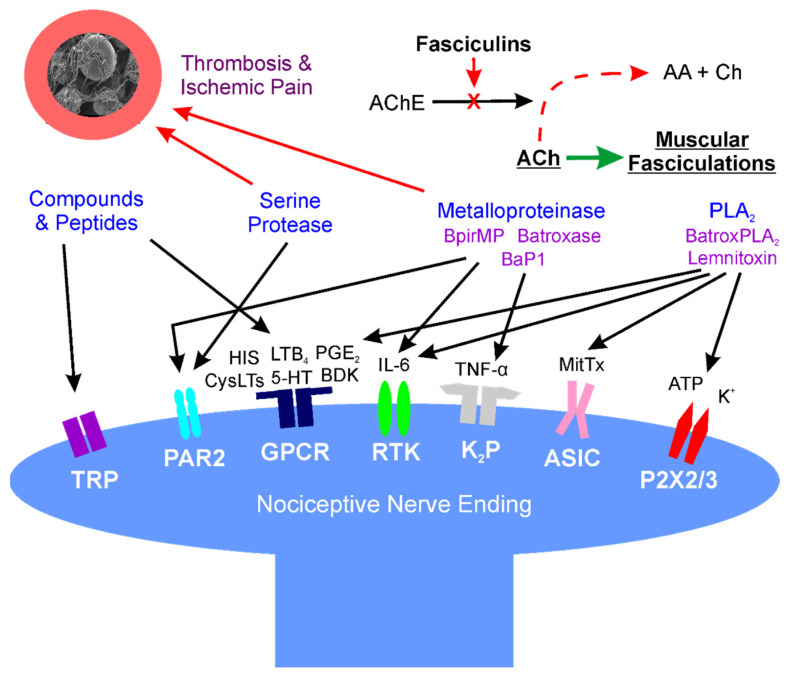
Location: molecular mechanisms of venom mediated pain. Diagram of interactions of snake venom compounds and proteins with nociceptive nerve endings and other key systems that result in pain. As explained in detail in the text, the indicated compounds and proteins activate receptors either directly or via products of enzymatic catalysis. Further, arterial thrombosis and ischemic pain remote from the bite are caused by serine proteases and metalloproteinases; also, at neuromuscular junctions distant from the bite, fasciculins inactivate acetylcholinesterase activity, allowing relentless activation of muscular activity via acetylcholine. AA—acetic acid; AChE—acetylcholinesterase; ASIC—acid sensing ion channel; ATP—adenosine triphosphate; BaP1, Batroxase, BpirMP—examples of metalloproteinases; BatroxPLA_2_, Lemnitoxin—examples of PLA_2_; BDK—bradykinin; Ch—choline; CysLTs, LTB_4_—examples of leukotrienes; GPCR—G-protein coupled receptor; HIS—histamine; IL-6—interleukin 6; K^+^—potassium; K_2_P—two-pore potassium channel; MitTx—a low activity PLA_2_ molecule bound to a with Kunitz-like protein that directly activates ASIC; P2X2/3—purinoceptors 2X2 and 2X3; PAR2—protease-activated receptor 2; PGE_2_—prostaglandin E_2_; RTK—receptor tyrosine kinase; and, TNF-α—tumor necrosis factor-α; TRP—transient receptor potential channel.

**Figure 2 ijms-23-02128-f002:**
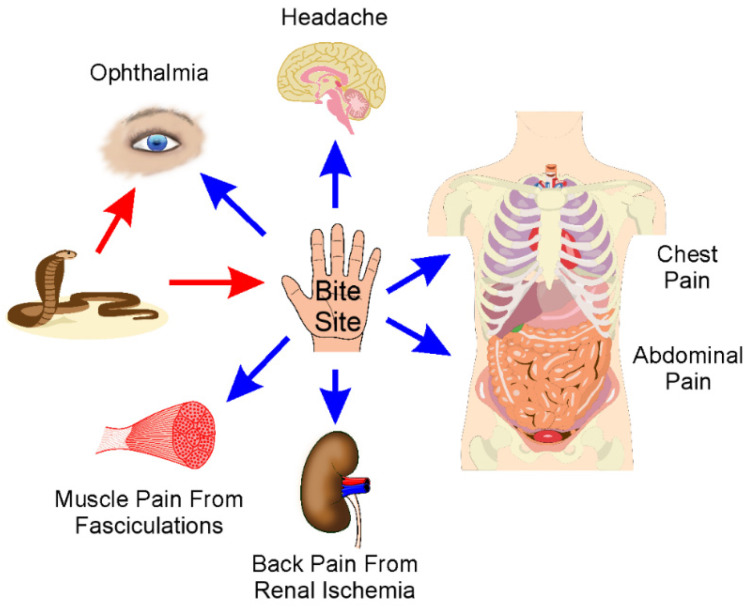
Location: sites of pain after envenomation. As described in detail in the text, snakes either spit venom or bite a body part (red arrows). Effects will occur locally or remotely by distribution from the bite site as indicated by the blue arrows. Subsequently, typical pain syndromes will occur as direct or indirect results of exposure to venom constituents or their enzymatic products.

**Figure 3 ijms-23-02128-f003:**
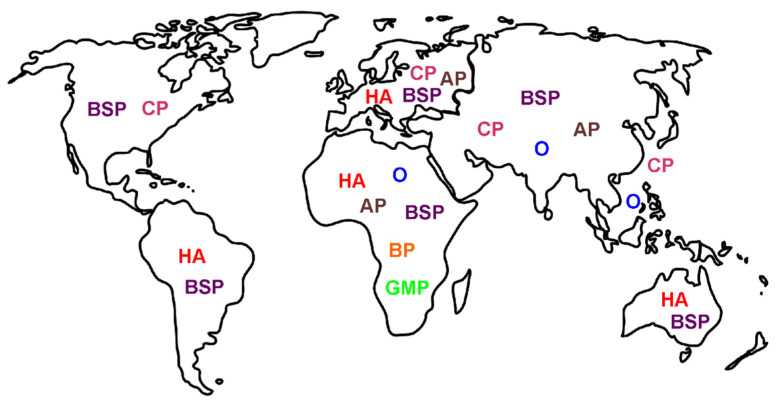
Location: nature of venom mediated pain as a function of geographic origin of the snake. As described in detail in the text, the geographic origin of the snake can dictate the pain syndrome observed. Typical pain syndromes reported to found in the indicated continents are represented by blue letters. Please note that the labeling is by continent in general and not a specific region. AP—abdominal pain (brown); BP—back pain (orange); BSP—bite site pain (purple); CP—chest pain (rose); GMP—generalized muscle pain (green); HA—headache (red); and O—ophthalmia (blue).

**Figure 4 ijms-23-02128-f004:**
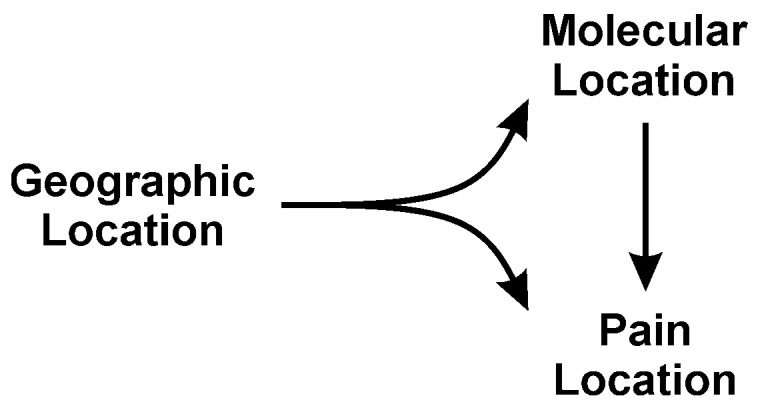
The paradigm of “location, location, location”.

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
