# Peer review of "Review of the Mechanisms of Snake Venom Induced Pain: It’s All about Location, Location, Location"

_ijms, 2022, doi:10.3390/ijms23042128_

Round 1

Reviewer 1 Report

The article "A Brief Review of the Molecular Mechanisms of Snake Venom Induced Pain: It's All About Location, Location, Location" by Nielsen & Wagner reviews the molecular mechanisms associated to the snake bite-induced pain syndromes. In the first part, the authors give a partial presentation of the components of snake venom that cause pain during envenomation. Then they present clinical cases illustrating the physiological localization of the pain felt after an envenomation. Finally, they recall that the great heterogeneity of snakes on the five continents is at the origin of venoms with numerous and different components, likely to act in a specific way on the mechanisms of pain.

Given its structure and title, I wonder whether this is a review or a scientific opinion.

For, although this is a short review, it lacks material, and in particular recent references in the molecular characterisation of pain-inducing venoms (in humans or animals). Indeed, pain is a consequence of the direct interaction, at the peripheral level, of venom components - or indirectly of substances released via these components - with the Ad and C type nociceptors. I recommend consulting the excellent review by Ferraz et al, 2019 which addresses this complex issue of pain induced by snake venoms, and their components (https://doi.org/10.3389/fevo.2019.00218) as well as the review by Jami et al, 2017 (https://doi.org/10.3390/toxins10010015). Finally, the issue of snake-induced pain is also addressed in the recent review by Gutiérrez et al (https://doi.org/10.1038/nrdp.2017.63). None of these three reviews appear in the article by Nielsen & Wagner.

This review needs to be completed and improved. In its current state, it cannot be published in IJMS.

In this respect, venom components contain toxins that target receptors specifically expressed in nociceptors, such as TRPV1 or ASIC channels (ASIC channels only appear in line 80). In particular, the venom of the saw-scaled viper Echis coloratus (https://doi.org/10.1016/j.bbagen.2017.01.004), which contains a protein toxin activating TRPV1, should be added and discussed.

Furthermore, the MiTx toxins from the venom of the Texas coral snake (Micrurus tener tener) in https://doi.org/10.1038/nature10607 are not cited as such in the article (L80). These are not PLA2 in the strict sense, but rather PLA2-like proteins possessing a "minimal PLA2 specific activity" as stated by the authors of this study (see Suppl. Fig.1). Consequently, the venom of Micrurus tener tener does not induce pain via PLA2 activity but by activation of ASIC channels by specific toxins. Section 2.3 of the review should therefore be corrected to take this into account. Figure 1 should also be revised in the light of this element, as it mentions the activation of ASIC channels by PLA2, which is only partially true.

Furthermore, other studies have investigated PLA2-mediated inflammatory pain in snake venoms (see for review https://doi.org/10.1007/978-94-007-6452-1_10). In particular, the effects of Crotalus durissus sp. venom are described in https://doi.org/10.1016/S0041-0101(03)00037-0. In addition, a similar mechanism is proposed for a PLA2 purified from Naja mocambique mocambique venom (https://doi.org/10.1097/MPA.0b013e3185d9b9b). More recent studies mention Lemnitoxin, from Micrurus lemniscatus coral snake venom, as a PLA2 with pro-inflammatory effects (https://doi.org/10.1016/j.toxlet.2016.06.005), or BatroxPLA2 from Bothrops atrox (https://doi.org/10.1016/j.molimm.2017.03.008).

Section 2.4 discusses pain induced by serine protease enzymes, mentioning the (inflammatory) mechanism rather briefly, but without any examples of snake venoms. The references here are general studies or on cancer-related pain... We must therefore propose snake venoms by citing the species and the serine protease enzymes of these venoms responsible for nociceptive inflammation. Recently, the P-I metalloprotease Batroxase from Bothrops atrox venom has been described as such a toxin (https://doi.org/10.1016/j.molimm.2017.03.008). Similarly, serine proteases from Bothrops jararaca venom are directly linked to hyperalgesic effects in vivo (https://doi.org/10.1016/j.toxicon.2009.07.025).

Similarly in section 2.5, no examples of snake venoms containing pro-inflammatory metalloproteinases are presented. However, there are numerous examples in the literature where this type of toxin is found, and the structure-function relationship described. Notably, the BaP1 and BpirMP toxins of Bothrops asper and Bothrops pirajai venoms induce hypernociception effects in vivo (https://doi.org/10.1038/sj.bjp.0707351 and https://doi.org/10.1016/j.molimm.2015.09.023). Batroxase, a metalloproteinase isolated from Bothrops atrox venom, is responsible for pro-inflammatory and hyperalgesic effects in vivo in rats (https://doi.org/10.1016/j.intimp.2015.06.001).

Section 2.6 is rather peculiar: pain related to fasciculins from mamba venoms is one of the undesirable effects, but little described in the literature, as it can be considered that all excitatory neurotoxins active at the neuromuscular junction are likely, through the activating effects they produce on muscle fibres, to produce painful effects. This section therefore needs to be better argued, with references to support the claim.

L121-126 do not rely on any references, but propose that “perhaps” remotely induced pain, e.g. visceral pain, is due to the release of small proalgesic molecules from venom enzymes. This hypothesis must be supported by results as the review is supposed to interrogate the sites of envenomation with the diffusion - local or global - of post-bite pain.

Similarly, L162-165 is not based on any reference, which shows that active molecules of a venom present in the oedematous fluid do not become circulating...

Considering what the authors propose to explain the pain experienced after an ophidian bite:

- Headache L192-193: “a great deal of investigation is needed to determine direct or indirect molecular mechanisms responsible”

- Chest pain L227-229: “multiple remote, direct and indirect molecular mechanisms are responsible for chest pain after envenomation, but further investigation will be required to further define them”.

- Abdominal pain L255-257 : “the precise molecular mechanisms responsible for severe generalized pain after snake bite remain to be defined”

Well, these observations are very partial, based on few clinical cases, and sometimes suggested by the authors.

To complete this review and considering that pain systematically comes from peripheral or central activation of nociceptors, the authors should add a figure representing a nociceptive sensory fibre, with its peripheral receptors and membrane channels and the different mechanisms that are likely to activate these neurons by snake venom toxins (directly or indirectly). This figure will help to illustrate sections 2 (venom molecules) and 3.2 (local pain). It will make the review clearer, especially at the molecular level (see Figure 2 in https://doi.org/10.1124/pr.111.005322 and Figure 4 in https://doi.org/10.1016/j.cell.2009.09.028).

A second figure also seems important to add: a representation of a human body, where all the sites concerned by the occurrence of pain would be noted (bite site, eye, back, thorax, etc.) and taking up the elements described in section 3.

Finally, it would be interesting to have a Figure with a geographical map showing the snakes in their endemic context and to associate the epidemiological studies carried out on these species to truly have a map making it possible to correlate snakes and envenomation/pain symptoms.

Figure 1 lacks precision.

The word 'perhaps' appears 4 times in this review. Rather than proposing ideas that are unsubstantiated, it is better to put forward arguments that are supported by published data. This would avoid leaving too much room for doubt, even if - as the authors mention - many questions are still unanswered. There is a lack of comprehensive epidemiological studies over a long period of time in this field.

Minor

L1-2: The title could be shortened to “Molecular Mechanisms of Snake Venom Induced Pain: It’s All About Location, Location, Location” and the type of paper should be classified as “short review” or even “opinion”.

L41: composed of biogenic amines, enzymes, toxins, peptides, and other substances to incapacitate

L42: Examples of such compounds include small molecular weight proteins

L160: “to be superior to non-specific treatments alone”. This is what I understood.

L306: Section 4 has only one subsection 4.1 . (and not 4.2, 4.3…).

L313: [36-36]

Author Response

“The article "A Brief Review of the Molecular Mechanisms of Snake Venom Induced Pain: It's All About Location, Location, Location" by Nielsen & Wagner reviews the molecular mechanisms associated to the snake bite-induced pain syndromes. In the first part, the authors give a partial presentation of the components of snake venom that cause pain during envenomation. Then they present clinical cases illustrating the physiological localization of the pain felt after an envenomation. Finally, they recall that the great heterogeneity of snakes on the five continents is at the origin of venoms with numerous and different components, likely to act in a specific way on the mechanisms of pain.”

“Given its structure and title, I wonder whether this is a review or a scientific opinion.”

            We reassure the reviewer that our work is a review.

“For, although this is a short review, it lacks material, and in particular recent references in the molecular characterisation of pain-inducing venoms (in humans or animals). Indeed, pain is a consequence of the direct interaction, at the peripheral level, of venom components - or indirectly of substances released via these components - with the Ad and C type nociceptors.”

            Before addressing this comment and all the subsequently presented comments, we wish to convey our thoughts to this reviewer. First, it is astounding that this reviewer took so much time to read our paper and offer suggestion – 15 manuscripts or articles by our count. The references added excellent material, and we very much agree with the suggestion to add multiple species, venoms, proteins and enzymes to our work. The reviewer clearly has a strong command of the subject, and we are grateful that so much time was taken. The vast majority of the suggestions were valid with only one exception wherein the reviewer was under the impression that one paper discussed hyperalgesia in a mouse model being serine protease mediated when instead it was mediated mostly by metalloproteinases. Nevertheless, this motivated us to successfully find a replacement manuscript for the indicated section of the paper. As for the aforementioned comment, we have made reference to the A-delta and C fibers in the text.

            In summary, thank you. We hope that you like what we have done with your fine and complete suggestions that included so many references.

“I recommend consulting the excellent review by Ferraz et al, 2019 which addresses this complex issue of pain induced by snake venoms, and their components (https://doi.org/10.3389/fevo.2019.00218)...”

            This reference has been added, and its details mentioned in the appropriate section of the text.

“…as well as the review by Jami et al, 2017 (https://doi.org/10.3390/toxins10010015).

            This reference has been added, and its details mentioned in the appropriate section of the text.

“Finally, the issue of snake-induced pain is also addressed in the recent review by Gutiérrez et al (https://doi.org/10.1038/nrdp.2017.63). None of these three reviews appear in the article by Nielsen & Wagner.”

            This reference has been added, and its details mentioned in the appropriate section of the text.

“This review needs to be completed and improved. In its current state, it cannot be published in IJMS.”

            We have made significant changes that we hope will satisfy the reviewer.

 “In this respect, venom components contain toxins that target receptors specifically expressed in nociceptors, such as TRPV1 or ASIC channels (ASIC channels only appear in line 80). In particular, the venom of the saw-scaled viper Echis coloratus (https://doi.org/10.1016/j.bbagen.2017.01.004), which contains a protein toxin activating TRPV1, should be added and discussed.”

            This reference has been added, and its details mentioned in the appropriate section of the text.

“Furthermore, the MiTx toxins from the venom of the Texas coral snake (Micrurus tener tener) in https://doi.org/10.1038/nature10607 are not cited as such in the article (L80). These are not PLA2 in the strict sense, but rather PLA2-like proteins possessing a "minimal PLA2 specific activity" as stated by the authors of this study (see Suppl. Fig.1). Consequently, the venom of Micrurus tener tener does not induce pain via PLA2 activity but by activation of ASIC channels by specific toxins. Section 2.3 of the review should therefore be corrected to take this into account.

            We appreciate this correction and have made it in our text.

“Figure 1 should also be revised in the light of this element, as it mentions the activation of ASIC channels by PLA2, which is only partially true.”

            This has been changed in the text. The new figure 1 is already complex, so this pathway is shown in general with reference to the specific details in text.

“Furthermore, other studies have investigated PLA2-mediated inflammatory pain in snake venoms (see for review https://doi.org/10.1007/978-94-007-6452-1_10).”

“In particular, the effects of Crotalus durissus sp. venom are described in https://doi.org/10.1016/S0041-0101(03)00037-0.”

             These references have been added, and their details mentioned in the appropriate section of the text.

“In addition, a similar mechanism is proposed for a PLA2 purified from Naja mocambique mocambique venom (https://doi.org/10.1097/MPA.0b013e3185d9b9b).”

            This reference has been added, and its details mentioned in the appropriate section of the text.

“More recent studies mention Lemnitoxin, from Micrurus lemniscatus coral snake venom, as a PLA2 with pro-inflammatory effects (https://doi.org/10.1016/j.toxlet.2016.06.005), or BatroxPLA2 from Bothrops atrox (https://doi.org/10.1016/j.molimm.2017.03.008).”

            These references have been added, and their details mentioned in the appropriate section of the text.

“Section 2.4 discusses pain induced by serine protease enzymes, mentioning the (inflammatory) mechanism rather briefly, but without any examples of snake venoms. The references here are general studies or on cancer-related pain... We must therefore propose snake venoms by citing the species and the serine protease enzymes of these venoms responsible for nociceptive inflammation. Recently, the P-I metalloprotease Batroxase from Bothrops atrox venom has been described as such a toxin (https://doi.org/10.1016/j.molimm.2017.03.008).”

            We whole-heartedly agree that specific references to specific snakes adds needed examples and details. This reference has been added, and its details mentioned in the appropriate section of the text.

“Similarly, serine proteases from Bothrops jararaca venom are directly linked to hyperalgesic effects in vivo (https://doi.org/10.1016/j.toxicon.2009.07.025).”

            We have included this reference, but it demonstrated that serine proteases had non-participation in nociception (figure 2). Nevertheless, we found an article that does implicate serine proteases purified from snake venom as pain causing agents: Menaldo, D.L.; Bernardes, C.P.; Pereira, J.C.; Silveira, D.S.; Mamede, C.C.; Stanziola, L.; Oliveira, Fd., Pereira-Crott, L.S.; Faccioli, L.H.; Sampaio, S.V. Effects of two serine proteases from Bothrops pirajai snake venom on the complement system and the inflammatory response. Int Immunopharmacol. 2013, 15, 764-771.

“Similarly in section 2.5, no examples of snake venoms containing pro-inflammatory metalloproteinases are presented. However, there are numerous examples in the literature where this type of toxin is found, and the structure-function relationship described. Notably, the BaP1 and BpirMP toxins of Bothrops asper and Bothrops pirajai venoms induce hypernociception effects in vivo (https://doi.org/10.1038/sj.bjp.0707351 and https://doi.org/10.1016/j.molimm.2015.09.023).”

            This reference has been added, and its details mentioned in the appropriate section of the text.

“Batroxase, a metalloproteinase isolated from Bothrops atrox venom, is responsible for pro-inflammatory and hyperalgesic effects in vivo in rats (https://doi.org/10.1016/j.intimp.2015.06.001).”

            This reference has been added, and its details mentioned in the appropriate section of the text.

“Section 2.6 is rather peculiar: pain related to fasciculins from mamba venoms is one of the undesirable effects, but little described in the literature, as it can be considered that all excitatory neurotoxins active at the neuromuscular junction are likely, through the activating effects they produce on muscle fibres, to produce painful effects. This section therefore needs to be better argued, with references to support the claim.”

            We have emphasized why the one reference mentions pain after mamba bite, and we have added a reference that describes the molecular action of fasciculins. If people have significant pain after one dose of succinylcholine, then hours of fasciculations would be terrible. We suspect that the reason why more pain is not reported is that the authors of the case reports are delighted their patients lived at all.

“L121-126 do not rely on any references, but propose that “perhaps” remotely induced pain, e.g. visceral pain, is due to the release of small proalgesic molecules from venom enzymes. This hypothesis must be supported by results as the review is supposed to interrogate the sites of envenomation with the diffusion - local or global - of post-bite pain.”

            We have removed the word “perhaps” from this statement. We did mention, explicitly, which types of pain are or are not relieved after antivenom administration. We are not sure what the reviewer means by interrogation – determining what particular enzymes directly inflict specific pain syndromes remains to be performed.

“Similarly, L162-165 is not based on any reference, which shows that active molecules of a venom present in the oedematous fluid do not become circulating...”

            In the preceding passages we did reference evidence that supported this statement. To further emphasize this, we have rephrased the sentence thusly: “From a molecular standpoint, once the venom induces edema, its enzymatic constituents are prevented from being adsorbed as evidenced by the lack of symptoms of systemic envenomation; and, conversely, the macromolecular antibodies of antivenom are not able to access the venom enzymes inflicting pain on the surface of the eye as documented by the lack of efficacy of antivenom administration.”

“Considering what the authors propose to explain the pain experienced after an ophidian bite:

- Headache L192-193: “a great deal of investigation is needed to determine direct or indirect molecular mechanisms responsible”

- Chest pain L227-229: “multiple remote, direct and indirect molecular mechanisms are responsible for chest pain after envenomation, but further investigation will be required to further define them”.

- Abdominal pain L255-257 : “the precise molecular mechanisms responsible for severe generalized pain after snake bite remain to be defined”

Well, these observations are very partial, based on few clinical cases, and sometimes suggested by the authors.”

            We can only provide to the readership what is known, and the purpose of any review is to identify critical gaps in knowledge. As best we can tell, no previous work has tried to integrate the molecular, anatomical/clinical and global/evolutionary aspects of snake bite pain. Thus, clearly identifying what remains to be investigated from the basic science standpoint and clinical pain syndrome standpoint is the explicit purpose of our work.        

 “To complete this review and considering that pain systematically comes from peripheral or central activation of nociceptors, the authors should add a figure representing a nociceptive sensory fibre, with its peripheral receptors and membrane channels and the different mechanisms that are likely to activate these neurons by snake venom toxins (directly or indirectly). This figure will help to illustrate sections 2 (venom molecules) and 3.2 (local pain). It will make the review clearer, especially at the molecular level (see Figure 2 in https://doi.org/10.1124/pr.111.005322 and Figure 4 in https://doi.org/10.1016/j.cell.2009.09.028).”

            We appreciate this suggestion and found the references provided by the reviewer very helpful. We hope that the reviewer finds the new figure 1 acceptable in this matter.

“A second figure also seems important to add: a representation of a human body, where all the sites concerned by the occurrence of pain would be noted (bite site, eye, back, thorax, etc.) and taking up the elements described in section 3.”

            We have created a Figure 2 as requested.

“Finally, it would be interesting to have a Figure with a geographical map showing the snakes in their endemic context and to associate the epidemiological studies carried out on these species to truly have a map making it possible to correlate snakes and envenomation/pain symptoms.”

            While this is an interesting concept, given the hundreds of species and variable pain syndromes, such a figure would be far too complex for the authors to manufacture or for the readership to comprehend. This level of detail is far beyond the scope of the present work. We have added figure 3 that indicates where some of the pain syndromes have been reported as per the text of the section concerning this matter. We also added some additional passages to address this matter calling for more detailed epidemiological studies that assess pain in the setting of snake bite.

“Figure 1 lacks precision.”

            Now we label this as Figure 4 and have modified it to simply present the paradigm of “location, location, location”.

“The word 'perhaps' appears 4 times in this review. Rather than proposing ideas that are unsubstantiated, it is better to put forward arguments that are supported by published data. This would avoid leaving too much room for doubt, even if - as the authors mention - many questions are still unanswered. There is a lack of comprehensive epidemiological studies over a long period of time in this field.”

            One of our main purposes in writing this review was to identify doubt and the vast unknown that needs to be pursued. We are left with at best inductive reasoning with the limited data available. The abatement of pain with antivenom is most likely secondary to inactivation of a protein, not neutralization of a small molecular compound.

Minor

“L1-2: The title could be shortened to “Molecular Mechanisms of Snake Venom Induced Pain: It’s All About Location, Location, Location” and the type of paper should be classified as “short review” or even “opinion”.”

            We have changed the title to accommodate both reviewers. If the reviewer wants the article labeled differently, we would note that given the number of references and scientific content, this article is still simply a review as per the classifications of the journal. As per the style of the journal, a manuscript is either an article or review.

“L41: composed of biogenic amines, enzymes, toxins, peptides, and other substances to incapacitate”

            We have added the missing comma.

“L42: Examples of such compounds include small molecular weight proteins”

            We removed the extra ‘include’.

“L160: “to be superior to non-specific treatments alone”. This is what I understood.”

            We are happy to revise this statement as the reviewer wishes.

“L306: Section 4 has only one subsection 4.1 . (and not 4.2, 4.3…).”

            We have removed the 4.1.

“L313: [36-36]”

            We replaced this with [36-39,41-46] prior to adding all the new references. We appreciate the reviewer finding this oversight.

Reviewer 2 Report

The aim of the article was to give a brief review of some mechanisms involved in the pain induced by snake venoms.

The review is really good, it is well written, and it is an important contribution to the area.

I have some few points to comment

  • My suggestion is to exclude the word “molecular” from then title. The article describes that little is really known about mechanisms involved in pain induced by venom snakes, which I agree. Some known mediators are described, some clinical cases were reported, some good suggestions were included, however, molecular mechanisms are not really described in the article, mainly because they are unknown. Then, my suggestion it to keep only “mechanisms” in the title
  • The review aims to describe which are the causes of each types of pain, focusing on local direct or indirect action and remote action of the substances present in the venom, especially proteins. I really liked how this description was done. However, in general, I believe that the consequence of the inflammation induced by the venoms and the contribution of these inflammatory mediators to pain is missing. The authors focused on the ability of metallo, serino and phospholipases to circulate or not. However, not only these enzymes contribute to systemic pain, but inflammatory mediators in general, such as interleukins, TNF, prostaglandins, formed as a consequence of these enzymes on the tissues, contribute to pain. And this part is absent in the article.

Some minor points

Line 42 – exclude “include”

Lines 57-58 – “This category includes compounds such as biogenic amines (e.g., histamine, serotonin), kinins, eicosanoids, and other peptides that bind to receptors of biogenic amines”. In fact, all these substances bind to their specific receptors and not to the receptors of biogenic amines. Consider to change the phrase to “This category includes compounds such as biogenic amines (e.g., histamine, serotonin), kinins, eicosanoids, and other peptides that bind to their specific receptors”.

Lines 69 – the meaning of the phrase is not understandable. I believe that the “that” in line 69 should be suppressed

Line 74 – exclude “these of”

Author Response

“The aim of the article was to give a brief review of some mechanisms involved in the pain induced by snake venoms. The review is really good, it is well written, and it is an important contribution to the area.”

            We appreciate the reviewer’s kind words and hope that the revision, with the thoughtful suggestions of both reviewers will further improve our work.

“I have some few points to comment

My suggestion is to exclude the word “molecular” from then title. The article describes that little is really known about mechanisms involved in pain induced by venom snakes, which I agree. Some known mediators are described, some clinical cases were reported, some good suggestions were included, however, molecular mechanisms are not really described in the article, mainly because they are unknown. Then, my suggestion it to keep only “mechanisms” in the title”

            We have changed the title as suggested by this and the other reviewer.

“The review aims to describe which are the causes of each types of pain, focusing on local direct or indirect action and remote action of the substances present in the venom, especially proteins. I really liked how this description was done. However, in general, I believe that the consequence of the inflammation induced by the venoms and the contribution of these inflammatory mediators to pain is missing. The authors focused on the ability of metallo, serino and phospholipases to circulate or not. However, not only these enzymes contribute to systemic pain, but inflammatory mediators in general, such as interleukins, TNF, prostaglandins, formed as a consequence of these enzymes on the tissues, contribute to pain. And this part is absent in the article.”

            The reviewer makes a fine point that we needed to further note in section 2.2. Indeed, even if peptides/enzymes seem to bother one organ system, that does not preclude release of mediators from that organ that affects the organism in general. We have added the following statement in 2.2: “However, as the subsequently described venom proteins are released into the circulation and cause pain in distant organs in a syndromic fashion, it should be remembered that as end-organ inflammation increases, so does release of the aforementioned small molecular weight compounds that may contribute to pain systemically.”

Some minor points

“Line 42 – exclude “include”
            We have corrected this mistake.

“Lines 57-58 – “This category includes compounds such as biogenic amines (e.g., histamine, serotonin), kinins, eicosanoids, and other peptides that bind to receptors of biogenic amines”. In fact, all these substances bind to their specific receptors and not to the receptors of biogenic amines. Consider to change the phrase to “This category includes compounds such as biogenic amines (e.g., histamine, serotonin), kinins, eicosanoids, and other peptides that bind to their specific receptors”.”

            We thank the reviewer for this correction and have changed the phrase accordingly.

“Lines 69 – the meaning of the phrase is not understandable. I believe that the “that” in line 69 should be suppressed”

            We again thank the reviewer for identifying this problem. The statement has been corrected.

“Line 74 – exclude “these of””

            The indicated change was made.

Round 2

Reviewer 1 Report

The authors of this review have significantly modified the draft, adding a large number of bibliographic references, and proposing several new figures. The text has also been modified and expanded. The review gains in clarity.

Regarding figure 1, rather than a simple diagram showing components of snake venoms and their molecular targets, I really think it would have been original to represent the peripheral terminal of a nociceptor, at the site of a venomous bite. And to add the components of the venom and their targets: I propose a schematic for a more illustrative figure (of course that needs to be improved). I believe that a review has the duty to illustrate in the clearest possible way the statements it conveys. I am convinced that an illustrative figure improves a review and increases the citations that will be made of it. The suggested figure can be improved and may serve as a template for a final figure.

In addition, in figure 1, fasciculins look a little isolated: I think that they should somehow be integrated into the figure (I understand that muscle contractions cause cramps that generate mechanical pain, but this has to be confirmed). Thrombosis and the resulting ischemic pain can probably be added to this figure as well.

For Figure 3, the authors propose a geographical map representing the location of the different pain syndromes associated with snakebites. They indicate that a detailed map would be too complicated to produce, given the very large number of cases of ophidian bites, for which there are hundreds of clinical descriptions. However, this map appears to be extremely simplified, perhaps too simplified. For instance, rattlesnake or coral snake bites in North America do not appear, yet there is reason to believe that they are responsible for painful envenomations. See doi: 10.1186/1471-227X-11-2, doi: 10.1053/j.ctsap.2006.10.005, PMID: 30085573 and an older review (doi:10.1001/archsurg.1960.01300050021004). Thus, the map can probably be improved.

Author Response

“The authors of this review have significantly modified the draft, adding a large number of bibliographic references, and proposing several new figures. The text has also been modified and expanded. The review gains in clarity.”

            We appreciate the reviewer’s kind words, and again thank him/her for the suggestions.

“Regarding figure 1, rather than a simple diagram showing components of snake venoms and their molecular targets, I really think it would have been original to represent the peripheral terminal of a nociceptor, at the site of a venomous bite. And to add the components of the venom and their targets: I propose a schematic for a more illustrative figure (of course that needs to be improved). I believe that a review has the duty to illustrate in the clearest possible way the statements it conveys. I am convinced that an illustrative figure improves a review and increases the citations that will be made of it. The suggested figure can be improved and may serve as a template for a final figure.”

            We appreciate this criticism and have once again tried to provide the reviewer with a figure that satisfies him/her. We are concerned that further complexity beyond what we have placed in the new illustration will provide less clarity, causing the reader to lose interest.

The figure is now in color, includes some of the examples of proteins and peptides that activate the indicated receptors, includes a scanning electron micrograph of a thrombus, and includes a more elaborate series of reactions that occur during envenomation by fasciculins.

“In addition, in figure 1, fasciculins look a little isolated: I think that they should somehow be integrated into the figure (I understand that muscle contractions cause cramps that generate mechanical pain, but this has to be confirmed).”

            We have further detailed this section of the figure, adding the products of AChE catalysis of acetylcholine. Just like thrombosis, fasciculations happen in a different location away from the nociceptor, so some isolation is necessary. We hope the reviewer will recognize this and will find the figure acceptable.

            We are dismayed that the reviewer is unsure that fasciculations can cause pain. We presented a case report of such pain after a mamba bite [84] and provided the biochemical mechanism [85]. We also provided data from the anesthesiology literature concerning muscle pain following the administration of the depolarizing neuromuscular blocking agent, succinylcholine [86]. It is unclear what remains to be confirmed.

            While we have been hesitant to be more assertive in this matter, at this point in the review there are a few matters that we would respectfully note. First, the senior and corresponding author is a professor of anesthesiology, with 35 years of clinical experience. He has seen his share of people administered succinylcholine complain of muscular pain in the postoperative period – especially those with more musculature. If this sort of problem happens with a few minutes of fasciculations after a medication, how much worse can it be after hours of disordered, chaotic fasciculations endured by patients bitten by mambas? Maybe not all may suffer, but it is very likely that many do experience muscular pain during recovery. Second, a simple search of PubMed with the phrase “succinylcholine and muscle pain” returned 304 manuscripts. Another search using the phrase “muscle fasciculations and pain” brought back 175 manuscripts. We do not want to make this a sticking point for the reviewer, but we are certain, based on our clinical experience with fasciculations and muscle pain, that it is appropriate to include this phenomenon. Please let us agree to allow future investigations to further define the incidence and severity of this pain syndrome.

            Again, as we indicated in our Conclusion, future investigation is needed to more methodically document pain syndromes such as post-fasciculation muscle pain in settings wherein the life of the patient is considered more central. We use the sentence “Documentation of specific pain syndromes in greater detail in future epidemiological studies of snake bite is also critical.” to further emphasize this point.

“Thrombosis and the resulting ischemic pain can probably be added to this figure as well.”

            We have kept this concept in the new figure and now add a scanning electron micrograph of a thrombus kindly provided to us by Professor Resia Pretorius of South Africa. We hope that this adds to the new figure 1 what the reviewer is looking for in a citable figure.

“For Figure 3, the authors propose a geographical map representing the location of the different pain syndromes associated with snakebites. They indicate that a detailed map would be too complicated to produce, given the very large number of cases of ophidian bites, for which there are hundreds of clinical descriptions. However, this map appears to be extremely simplified, perhaps too simplified. For instance, rattlesnake or coral snake bites in North America do not appear, yet there is reason to believe that they are responsible for painful envenomations. See doi: 10.1186/1471-227X-11-2, doi: 10.1053/j.ctsap.2006.10.005, PMID: 30085573 and an older review (doi:10.1001/archsurg.1960.01300050021004). Thus, the map can probably be improved.”

            While we appreciate the reviewer’s comments, it is nevertheless impossible to create a map that meets his original expectations. We did not include any of the aforementioned suggested articles, as we already have cited more manuscripts than would be needed to further fill out types of pain in Figure 3. We have now added essentially all of the pain syndromes, color-coded them, and have added text to address this matter. Our fourth reference mentions the numerous species alluded to by the author in North America, Central America, South America, and the remaining continents. We hope that his final modification of figure 3 is sufficient, as we believe further modification or added complexity will result in a figure that will not be easily understood or appreciated by the readership.

Round 3

Reviewer 1 Report

The draft has undergone many modifications, and I now consider that this review can be published as is.